# Microwave Synthesis of Visible-Light-Activated g-C_3_N_4_/TiO_2_ Photocatalysts

**DOI:** 10.3390/nano13061090

**Published:** 2023-03-17

**Authors:** Maria Leonor Matias, Ana S. Reis-Machado, Joana Rodrigues, Tomás Calmeiro, Jonas Deuermeier, Ana Pimentel, Elvira Fortunato, Rodrigo Martins, Daniela Nunes

**Affiliations:** 1CENIMAT|i3N, Department of Materials Science, School of Science and Technology, NOVA University Lisbon and CEMOP/UNINOVA, 2829-516 Caparica, Portugal; 2LAQV-REQUIMTE, Department of Chemistry, NOVA School of Science and Technology, Universidade NOVA de Lisboa, Campus de Caparica, 2829-516 Caparica, Portugal; 3Physics Department & I3N, Aveiro University, Campus Universitário de Santiago, 3810-193 Aveiro, Portugal

**Keywords:** g-C_3_N_4_/TiO_2_, microwave synthesis, heterostructures, photocatalysis, pollutant degradation

## Abstract

The preparation of visible-light-driven photocatalysts has become highly appealing for environmental remediation through simple, fast and green chemical methods. The current study reports the synthesis and characterization of graphitic carbon nitride/titanium dioxide (g-C_3_N_4_/TiO_2_) heterostructures through a fast (1 h) and simple microwave-assisted approach. Different g-C_3_N_4_ amounts mixed with TiO_2_ (15, 30 and 45 wt. %) were investigated for the photocatalytic degradation of a recalcitrant azo dye (methyl orange (MO)) under solar simulating light. X-ray diffraction (XRD) revealed the anatase TiO_2_ phase for the pure material and all heterostructures produced. Scanning electron microscopy (SEM) showed that by increasing the amount of g-C_3_N_4_ in the synthesis, large TiO_2_ aggregates composed of irregularly shaped particles were disintegrated and resulted in smaller ones, composing a film that covered the g-C_3_N_4_ nanosheets. Scanning transmission electron microscopy (STEM) analyses confirmed the existence of an effective interface between a g-C_3_N_4_ nanosheet and a TiO_2_ nanocrystal. X-ray photoelectron spectroscopy (XPS) evidenced no chemical alterations to both g-C_3_N_4_ and TiO_2_ at the heterostructure. The visible-light absorption shift was indicated by the red shift in the absorption onset through the ultraviolet-visible (UV-VIS) absorption spectra. The 30 wt. % of g-C_3_N_4_/TiO_2_ heterostructure showed the best photocatalytic performance, with a MO dye degradation of 85% in 4 h, corresponding to an enhanced efficiency of almost 2 and 10 times greater than that of pure TiO_2_ and g-C_3_N_4_ nanosheets, respectively. Superoxide radical species were found to be the most active radical species in the MO photodegradation process. The creation of a type-II heterostructure is highly suggested due to the negligible participation of hydroxyl radical species in the photodegradation process. The superior photocatalytic activity was attributed to the synergy of g-C_3_N_4_ and TiO_2_ materials.

## 1. Introduction

The earth’s surface is composed of 71% of water, with only 1% suitable for human consumption [1]. Not only is potable water scarce, but several factors, such as climate change, population growth, development of industrialization and water misuse, have also increased the production of atmospheric pollution [2] and have led to a deterioration of the available water in terms of quantity (e.g., over-exploitation and droughts) and also quality (e.g., eutrophication) [3,4,5]. According to the WHO (World Health Organization, 2022 [6]), over 2 billion people live in water-stressed countries, and millions of people die of severe diseases due to contaminated water and poor sanitation [6,7].

Dyes from textile industries significantly contribute to the water contamination problem [8], and in this regard, it is predicted that around 700,000 tonnes of dyes is produced annually worldwide [1], where 10–15% of that production ends up being discharged into the environment [9], threatening the human health and the ecosystem [10]. Consequently, the scientific output on the topic of dye removal has grown over the last few years [10]. Dyes can be divided into different groups according to their chemical structure, type of application [11], and they can also be separated into anionic, cationic and non-ionic dyes [11]. The largest and most important group of organic dyes is the azo dyes class [9]. This class of dyes is extensively used in many sectors, such as the textile, food, pharmaceutical and cosmetic industries [12]. Among azo dyes, methyl orange is one of the most stable. It is commonly used in the textile industry and as an acid-base indicator. Simultaneously, its environmental impact is of great concern, since it is resistant to biodegradation and can potentially induce severe consequences for animals and humans, such as gene mutations and cancer [12,13,14].

Different methods can be employed for the removal of textile dyes from effluents, including physical, chemical and biological treatments. Physical methods (such as adsorption and membrane filtration) have proved to be efficient in the treatment of industrial dye effluents. However, their major drawback is the increased sludge formation volume. Moreover, filtration methods, for instance, typically require high maintenance costs, which does not make them cost effective [11]. Biological treatments are generally considered an inexpensive and eco-friendly alternative to physical methods, consisting of dye degradation through the metabolic pathways or adsorption of living/dead biomass [15]. Nevertheless, these treatments fail to fully mineralize dyes if more complex compounds are present [10]. However, advanced oxidation processes (AOPs) have emerged as promising choices for the complete mineralization of organic contaminants. In particular, photocatalysis, a process included in the AOPs, allows the generation of hydroxyl radicals by using solar energy for complete mineralization of hazardous organic chemicals into carbon dioxide, water and mineral acids [16,17]. Additionally, photocatalytic experiments can be performed at room temperature (RT) [18] and do not generate secondary pollution [19], which makes this technique highly attractive for worldwide water remediation [20].

Lately, a lot of attention has been paid to semiconductors with visible-light absorption as photocatalysts due to the abundance of visible light on the solar spectrum [16,21,22,23,24]. Although titanium dioxide (TiO_2_) is a widely employed semiconductor in photocatalysis owing to its low toxicity, reduced cost and excellent photochemical stability [25], it mainly absorbs in the UV region [22], limiting its use in photocatalytic applications [16]. TiO_2_ naturally has three polymorphs at RT: anatase, rutile (tetragonal structures) and brookite (orthorhombic structure). The rutile phase is the most stable, whereas anatase and brookite phases are metastable and are easily converted to the rutile phase with temperatures higher than 600 °C [16,25,26,27]. Generally, anatase and rutile phases are the most studied in photocatalysis due to the difficulty in obtaining brookite [25,28]. TiO_2_ is an n-type semiconductor, displaying optical band gap energies around 3.0 eV, 3.2 eV and values from 3.13 to 3.40 eV for rutile, anatase and brookite phases, respectively [29,30]. However, these values might suffer variations arising from the presence of impurities or doping and depend on the synthesis’ approach [31,32].

Several techniques have already been studied to improve the photocatalytic performance of TiO_2_ under visible light, such as doping with metals [33]/non-metals [34], modification of surface defects [35] and the fabrication of heterostructures [21,36]. Among them, the formation of an enhanced heterostructure was proved not only to suppress the photogenerated charge carriers’ recombination rate but also to extend visible-light absorption, thereby improving the photocatalytic activity under visible light [21,37]. One material, which has started to be used to produce visible-light-activated photocatalysts based on heterostructures with TiO_2_, is graphitic carbon nitride [27,38,39].

Graphitic carbon nitride (g-C_3_N_4_) is a polymeric material with electron-rich properties, basic surface functionalities and H-bonding motifs, which make it suitable for catalytic applications [40]. In terms of the basic structure, this material is composed of layers of tri-s-triazine ring structures, connected by N atoms, wherein van der Waals forces hold its layered stacking to form a 2D π-conjugated polymeric network [41,42,43,44]. Moreover, it presents high thermal stability (up to 600 °C in air) and hydrothermal stability (it is insoluble in acidic, neutral or basic medium) [40]. This material is generally produced by thermal polycondensation of low-cost precursors, between 500 and 600 °C in air or inert atmosphere [45], involving carbon- and nitrogen-rich organic compounds [46], such as dicyandiamide [47], melamine [48,49], urea [49,50] and thiourea [49,51]. g-C_3_N_4_ presents reduced toxicity [52]; it is composed of two abundant earth elements (C and N), and hence, it is not considered a critical raw material [53]. It has a low band gap energy, between 2.7 and 2.8 eV, corresponding to a wavelength of 450–460 nm, which allows the absorption of visible light [54,55,56]. Furthermore, the photoreduction capability of this material is strongly promoted by its high redox potential of −1.3 V (vs. normal hydrogen electrode (NHE) at pH = 7 compared to the potential of −0.5 V for pure TiO_2_). However, its top energy level of the valence band (VB) is 1.4 V (vs. NHE at pH = 7 compared to 2.7 V for pure TiO_2_). Consequently, it has weak oxidative capability for water oxidation, resulting in insufficient hydroxyl radicals’ production [54]. Furthermore, the hybridization of the N 2p and C 2p states in the conduction band (CB) results in a fast recombination rate of photogenerated charge carriers. Moreover, this material is difficult to separate from water, may cause secondary pollution due to its dissolution and has low specific surface area, which restricts its use in photocatalytic applications. Among the possible strategies to tackle these issues, effective coupling of g-C_3_N_4_ with other semiconductors has previously shown to be advantageous in charge separation and improved absorption in the visible region [57,58,59,60].

A summary of the photocatalytic activities in the degradation of MO with different g-C_3_N_4_/TiO_2_ nanostructures can be seen in Table 1.

Several approaches have already been used to fabricate g-C_3_N_4_/TiO_2_ heterostructures, such as sol-gel [65], hydrothermal [58]/solvothermal syntheses [66], wet impregnation [67] and microwave-assisted methods [68,69]. Apart from the microwave irradiation process, these methods typically require high temperatures and/or long reaction times, hence being energy consuming [58,66,67,70,71]. Taking into account the minimization of energy, microwave irradiation appears as a good alternative for the synthesis of complex inorganic materials, since higher reaction yields can be obtained in such a short amount of time. Additionally, the method is simple, energy efficient, eco-friendly, offers uniform and selective heating (electromagnetic radiation is directly transferred to the reactive species, resulting in a localized superheating of the material [72]), and the selectivity of reactions can be improved [26,68,73,74].

To the best of our knowledge, no reports have yet demonstrated a simple and fast microwave-assisted approach to produce an enhanced g-C_3_N_4_/TiO_2_ heterostructure for the degradation of MO under visible light. From this viewpoint, this paper reports the fabrication of g-C_3_N_4_/TiO_2_ heterostructures through a simple, seed-layer-free, cost-effective and fast microwave-assisted approach, together with the evaluation of their photocatalytic activity. The structural, optical and electrochemical characterizations were performed by XRD, SEM and STEM, all equipped with energy-dispersive X-ray spectroscopy (EDS) detectors, atomic force microscopy (AFM), XPS, UV-VIS absorption and photoluminescence (PL) spectroscopies, and Mott–Schottky plots. The photocatalytic performance of different g-C_3_N_4_ amounts in TiO_2_ was investigated for the degradation of MO using a solar simulator for up to 240 min (4 h). For a better understanding of the species involved in the photocatalytic degradation process, experiments were performed with reactive oxygen species scavengers. Reusability tests were also carried out up to five consecutive cycles.

## 2. Experimental Procedure

### 2.1. Synthesis of g-C_3_N_4_ Nanopowder

g-C_3_N_4_ in powder form was directly prepared by calcination of 20 g of urea (from Sigma-Aldrich CAS: 57-13-6), which was transferred into a ceramic crucible and heated in a muffle furnace at 550 °C for 2 h, with a heating ramp of 35 min. The crucible was sealed to avoid nanopowder loss. The reaction was carried out with the exhaustion system turned on, since the thermal decomposition of urea releases toxic gases and vapors (such as ammonia) [75]. In the end, a pale-yellow powdered product was obtained.

### 2.2. Microwave Synthesis of the g-C_3_N_4_/TiO_2_ Heterostructures

For the synthesis of the g-C_3_N_4_/TiO_2_ heterostructures, the obtained g-C_3_N_4_ nanopowder was dispersed in 23 mL of ethanol (96%), and the solution was ultrasonically dispersed for 15 min to achieve homogeneity. Afterward, 0.3 mL of hydrochloric acid (HCl, 37% purity from Merck, Germany) was added, followed by 0.8 mL of dropwise Titanium (IV) isopropoxide (TTIP, 97% purity from Sigma-Aldrich, St. Louis, MO, USA (CAS: 546-68-9)). The solution was left to stir for 10 min. A volume of 20 mL of the prepared solution was then transferred to a 35 mL Pyrex vessel, which was placed in a CEM Discovery SP microwave (from CEM, Matthews, NC, USA). The synthesis was carried out for 1 h at 150 °C with a maximum power of 100 W and a maximum pressure of 250 psi. The resulting g-C_3_N_4_/TiO_2_ nanopowders were washed alternately with deionized water and isopropyl alcohol (IPA) several times using a centrifuge at 5320 rpm for 5 min each time and dried in a desiccator at 80 °C, under vacuum. Pure TiO_2_ was also obtained by performing a microwave synthesis under the same experimental conditions, except for the addition of g-C_3_N_4_ nanopowder. The materials produced will hereafter be called 15-GCN-T, 30-GCN-T and 45-GCN-T for the 15, 30 and 45 weight (wt.) % of g-C_3_N_4_ in TiO_2_, respectively. A schematic of the experimental procedure for the synthesis of g-C_3_N_4_/TiO_2_ heterostructures by microwave is visible in Figure 1.

### 2.3. Structural and Optical Characterization of the Nanostructures

XRD experiments were performed using a PANalytical’s X’Pert PRO MPD diffractometer (Almelo, The Netherlands) equipped with an X’Celerator detector and using CuKα radiation (λ = 1.540598 Å). XRD data were recorded from 10° to 90° 2θ range with a step of 0.05° in the Bragg–Brentano configuration. The simulated TiO_2_ brookite phase was indexed using the Inorganic Crystal Structure Database (ICSD) file No. 36408, while the simulated TiO_2_ rutile phase corresponded to ICSD file No. 9161 and the simulated TiO_2_ anatase phase to ICSD file No. 9852.

SEM images of the nanopowders were acquired with a Hitachi Regulus 8220 Scanning Electron Microscope (Mito, Japan) equipped with an Oxford EDS detector.

STEM, including high-angle annular dark-field (HAADF) imaging, and transmission electron microscopy (TEM) observations were carried out with a Hitachi HF5000 field-emission transmission electron microscope operated at 200 kV (Mito, Japan). This is a cold FEG TEM/STEM with a spherical aberration corrector for the probe, and it is equipped with one 100 mm^2^ EDS detector from Oxford Instruments. A drop of the sonicated dispersion was deposited onto lacey-carbon copper grids and allowed to dry before observation. The average particle size and standard deviation of the TiO_2_ nanostructures were calculated based on the SEM and TEM images obtained from the dimensions of 40 particles using ImageJ software [76].

AFM images were acquired with an Asylum Research MFP-3D Standalone system (Oxford Instruments, Abingdon, UK) operated under ambient conditions, in alternate contact mode, using commercially available silicon probes (Olympus AC160TS, f_0_ = 300 kHz, k = 26 N/m; Olympus Corporation, Tokyo, Japan). The images and height profiles were exported using Asylum Research’s software packages, after low-level plane fitting.

XPS measurements were performed with a Kratos Axis Supra (Kratos Analytical, Manchester, UK), using monochromated Al Kα irradiation (1486.6 eV). Detailed scans were acquired with an X-ray power of 120 W and a pass energy of 10 eV. The powder samples were mounted in an aluminum crucible and showed charge accumulation during the measurement. Charge neutralization with an electron flood gun was employed, and the lowest binding energy component of C 1s was referenced at 284.8 eV in CasaXPS, which was the software used for data analysis.

RT PL measurements were performed using a PerkinElmer LS55 luminescence spectrometer (PerkinElmer, Waltham, MA, USA) equipped with a Xenon lamp as an excitation source. The PL data were acquired from 400 to 600 nm, using an excitation wavelength of 350 nm.

Total reflectance and absorbance measurements were recorded by using a double-beam UV-VIS-NIR Shimadzu spectrophotometer with an integrating sphere in the range of 280–800 nm. The specular reflectance was also recorded in the same range to obtain the diffuse reflectance data (DRS) and estimate the band gap energies (E_g_). BaSO_4_ white powder was used as the reference. Measurements were carried out at RT.

### 2.4. Characterization of the Nanostructures as Photocatalysts in the Degradation of MO under Solar Simulating Light

The photocatalytic activities of pure TiO_2_, g-C_3_N_4_ and heterostructures 15, 30 and 45-GCN-T (in powder form) were evaluated at RT, considering the degradation of MO (C_14_H_14_N_3_NaO_3_S) from Sigma-Aldrich under a solar light simulating source. For each experiment, 25 mg of each nanopowder was dispersed with 50 mL of the MO solution (12.5 mg/L) and stirred for 30 min in the dark to establish the absorption–desorption equilibrium. Solar light exposure was conducted by using a WAVELABS LS-2 LED solar simulator (Germany) with AM 1.5 spectrum, at an intensity of 100 mW/cm^2^. The experiments were conducted under low constant magnetic agitation at 130 rpm. Absorption spectra were recorded using a PerkinElmer double-beam LAMBDA 365+ UV/Vis Spectrometer (Waltham, MA, USA), with different time intervals, up to a total of 240 min (4 h). The measurements were performed in the 300–600 nm range. Photolysis was also investigated by irradiating the MO solution without any photocatalyst.

For the reusability experiments, the catalysts were recovered by centrifugation at 6000 rpm for 5 min with further discarding of the supernatant. The nanopowders were then dried at 60 °C for 12 h, prior to the next exposure. Afterward, the recovered nanopowders were poured into a fresh solution and exposed to solar light under the same exposure times. Additional parallel photocatalytic experiments were performed to guarantee the same photocatalyst mass in each cycle (in this case, 25 mg).

To investigate the main active species involved in the photocatalytic degradation of MO, several reactive oxygen species (ROS) scavengers were added to the methyl orange solution prior to the addition of the catalyst. To evaluate the role of these species, ethylene diamine tetra acetic acid (EDTA, C_10_H_16_N_2_O_8_, ≥98% purity from Sigma-Aldrich, CAS: 60-00-4), isopropanol (IPA, C_3_H_8_O, 99.8% purity from Sigma-Aldrich) and *p*-benzoquinone (BQ, C_6_H_4_O_2_, ≥98% purity from Midland Scientific (Sigma-Aldrich), CAS: 106-51-4) were used, respectively, as hole scavengers (h^+^), hydroxyl radical scavengers (·OH) and superoxide radical scavengers (·O_2_^−^) [21,77]. The trapping experiments were conducted under the same conditions as those to evaluate the photocatalytic performance, additionally including 5 mL of a 0.5 mM aqueous solution of each scavenger [78]. A solution in the presence of the photocatalyst and 5 mL of deionized water (without scavenger) was also irradiated for comparison.

### 2.5. Electrochemical Characterization

To evaluate the electrochemical performance of the produced materials, a solution was prepared with 2.5 mg of each nanopowder, 10 μL of Nafion perfluorinated resin (5 wt. % in lower aliphatic alcohols and water, containing 15–20% of water content, from Sigma-Aldrich, CAS: 31175-20-9) and 0.25 mL of DMF (N,N-dimethylformamide, with ≥99% purity from Sigma-Aldrich). Then, indium tin oxide (ITO) glass samples were cut with dimensions of 2 cm ×  1 cm, and an appropriate amount of the solution was evenly coated on a sample area of around 1 cm^2^. Each layer was left to dry at 50 °C before applying a new layer. The photoelectrochemical properties were assessed on a standard three-electrode system (Gamry Reference 600, Warminster, PA, USA) by using a potentiostat model 600 from Gamry Instruments, Inc., Warminster, PA, USA and with an electrochemical interface (Gamry Echem Analyst). ITO glass coated with the photocatalyst, platinum wire and the standard electrode Ag/AgCl (3 M KCL) were used as the working electrode, the counter electrode and the reference electrode, respectively. The electrolyte was a solution with 0.5 M of Na_2_SO_4_ at pH = 6.3. The Mott–Schottky curves were measured at 1 kHz. The potentials measured against the (Ag/AgCl) reference were converted into normal hydrogen electrode (NHE) potentials, following Equation (1) [79]:(1)$$E_{\mathrm{FB}\;\left(\mathrm{NHE}\right)}=E_{\mathrm{Ag}/\mathrm{AgCl}}+E_{\mathrm{Ag}/\mathrm{AgCl}}^{{}^{\circ}}\;\left(E_{\mathrm{Ag}/\mathrm{AgCl}}^{{}^{\circ}}=0.1976\;V\;\mathrm{vs}.\;\mathrm{NHE}\;\mathrm{at}\;25{}^{\circ}C\right)$$

## 3. Results and Discussion

### 3.1. Structural Characterization

#### 3.1.1. XRD

The produced nanopowders, i.e., pure TiO_2_, pure g-C_3_N_4_, 15-GCN-T, 30-GCN-T and 45-GCN-T, were investigated by XRD. As seen in Figure 2, after heat treating urea at 550 °C for 2 h, two diffraction maxima at 13.2° and 27° (2θ) appear, indexed to the planes (100) and (002) of g-C_3_N_4_, respectively [80]. According to previous studies, the broad diffraction maximum detected at 13.2° is ascribed to the in-planar structure of repeated N-bridged tri-s-triazine units, whereas the maximum at 27° is attributed to the stacking of the conjugated aromatic systems. A slight shift could be observed in the diffractogram of the produced g-C_3_N_4_ nanopowder compared to the database, and for that reason, the simulated g-C_3_N_4_ was not presented. Nevertheless, the XRD data are consistent with the literature for this material [80,81,82,83]. For the pure TiO_2_ nanopowder, the experimental XRD diffractograms could be fully ascribed to the anatase TiO_2_ phase (ICSD file No. 9852). When it comes to the heterostructures, for all conditions, the TiO_2_ anatase diffractograms are also present. On the other hand, when g-C_3_N_4_ content is increased with respect to TiO_2_, a diffraction maximum at 27° appears, indicating the presence of graphitic carbon nitride. This can be observed in the 30-GCN-T and 45-GCN-T nanopowders (identified with orange arrows in Figure 2), confirming the co-existence of both materials (TiO_2_ and g-C_3_N_4_). The absence of graphitic carbon nitride patterns in the 15-GCN-T nanopowder is likely due to the low percentage present in this material. No further XRD maxima or impurities were detected in all the materials produced.

#### 3.1.2. Electron Microscopy

Figure 3 displays the SEM images of TiO_2_, g-C_3_N_4_ and g-C_3_N_4_/TiO_2_ heterostructures. As can be seen in Figure 3a, the microwave synthesis in the presence of ethanol as solvent resulted in several irregularly shaped particles with an average particle size of 543 ± 135 nm (Figure 3b). It is clearly seen in Figure 3b that the larger particles are composed of aggregates of TiO_2_ nanocrystals. A hollow sphere is evidenced on the inset of Figure 3b. The SEM images of g-C_3_N_4_ are shown in Figure 3c,d. Graphitic carbon nitride produced by direct calcination resulted in a two-dimensional (2D) structure composed of a stack of thin sheets with wrinkles and irregular shapes. Some micro-holes are also perceptible at the surface of the sheets, which were probably formed due to the escape of gases, such as NH_3_, during the high-temperature synthesis of g-C_3_N_4_. Consequently, the gas release could have etched the s-triazine network structure [84]. Figure 3e–j show the SEM images of the g-C_3_N_4_/TiO_2_ heterostructures. The similar shape and size of the TiO_2_ particles observed previously in Figure 3a are present in the 15-GCN-T material, covering some areas of the g-C_3_N_4_ sheets. A TiO_2_ film was also formed at the surface of the sheets (Figure 3f). Nevertheless, this film did not completely cover the g-C_3_N_4_ sheets for the 15-GCN-T material. Interestingly, when g-C_3_N_4_ content was further increased, as shown in Figure 3g–j, the larger TiO_2_ particles disintegrated, and smaller TiO_2_ agglomerates were formed, composing a film that was expressively thick for the 45-GCN-T material (Figure 3j). The close contact between the two materials is thus essential to improve the charge separation and strengthen the photocatalytic activity [85].

Due to the high amount of g-C_3_N_4_ in the 45-GCN-T material, some g-C_3_N_4_ nanosheets were not covered by TiO_2_, which could result in a faster recombination rate of photogenerated charge carriers, and lastly, a decrease in photocatalytic activity. For this reason, the 30-GCN-T material was selected for further investigation.

The 30-GCN-T material was investigated using STEM (Figure 4). In both STEM and bright-field TEM images (Figure 4a–d and Appendix A), the presence of a thin 2D nanostructure with a sheet-like structure is clear. The TiO_2_ nanostructures are also clearly discernible in the STEM and TEM images (Figure 4a–g). The ring diffraction pattern in Figure 4d attested that these particles are solely in the anatase TiO_2_ phase.

In accordance with the SEM results, it can be observed that the g-C_3_N_4_ sheets are covered with TiO_2_ nanostructures—however, not forming a continuous film—and larger agglomerates are also detected (Figure 4a–f) but, as observed previously, are expressively smaller than those for the pure TiO_2_ material. As reported in an analogous study [86], it can be seen that these agglomerates are formed by very fine TiO_2_ nanoparticles with irregular shapes, where near spherical nanocrystals and more elongated ones can be observed (Figure 4g–i). The average TiO_2_ particle size was found to be 5.17 ± 1.37 nm, and it can be seen from the particle size distribution (inset of Figure 4d) that smaller particles in the range of 4–7 nm are more likely to be found.

The STEM images (Figure 4g–i) confirm the close contact between the g-C_3_N_4_ sheets and the TiO_2_ nanocrystals, where the in-depth analysis conducted using detailed HAADF imaging (Figure 4i) revealed a clear interface between the TiO_2_ nanocrystal and the g-C_3_N_4_ sheet. As reported earlier in the literature [87], low crystallinity of g-C_3_N_4_ sheets was detected in the TEM measurements. From the atomic-resolution HAADF-STEM image in Figure 4i, the atomic columns are clearly visible, and given the Z-contrast, it can be assumed that the visible spots must correspond to Ti atoms [88]. These Ti atomic columns are perpendicular to each other, and a lattice spacing of ≈ 0.189 nm was measured, which perfectly matches the (200) and (020) atomic planes of anatase [89]. Two other distinct TiO_2_ nanocrystals were investigated in Figure 5a,b, revealing the (100) and (010) atomic planes of anatase with a lattice spacing of ≈ 0.378 nm [88]. The insets presented in Figure 4i and Figure 5a,b represent the fast Fourier transformation (FFT) images generated from areas indicated as A, B and C, respectively. As observed in the [001] zone axis, it is evident from the FFT patterns that the angles between (200) and (020) or between (100) and (010) are 90°, in accordance with the theoretical value reported for pure crystalline anatase TiO_2_ (ICSD file No. 9852). From the atomic-resolution HAADF images and FFT patterns, the tetragonal atomic arrangement on the (001) surface of the TiO_2_ nanocrystals can be inferred [89].

Energy-dispersive X-ray spectroscopy analyses were also carried out for the 30-GCN-T material. Figure 6a shows a magnified SE-STEM image of a g-C_3_N_4_ sheet with TiO_2_ agglomerates on its surface. From the EDS analysis, it is confirmed that the sheet is mainly composed of C (Figure 6b), N (Figure 6c) and O (Figure 6d), while Ti is present on the agglomerates, revealing a uniform distribution of this element (Figure 6e).

#### 3.1.3. AFM

AFM measurements were performed to determine the average height/thickness of the produced g-C_3_N_4_ nanosheets. The average thickness of g-C_3_N_4_ nanosheets was calculated based on several scans, which were taken from AFM images, and the value was found to be around 4 nm. One of those scans is depicted in Figure 7. Thin g-C_3_N_4_ nanosheets were also inferred from STEM analysis (Figure 4), where transparent nanosheets were observed. In accordance with previous studies, similar size thicknesses ranging from 2 to 15 nm were reported for g-C_3_N_4_ nanosheets [90,91,92].

#### 3.1.4. XPS

The surface chemical compositions of TiO_2_, g-C_3_N_4_ and 30-GCN-T materials were analyzed by XPS. Figure 8a shows the survey spectra of TiO_2_, g-C_3_N_4_ and 30-GCN-T materials, revealing the existence of C and N elements in the g-C_3_N_4_ nanopowder. Through the analysis of the survey spectra, the characteristic peaks of Ti and O appear in TiO_2_ along with Ti, O, N and C elements in 30-GCN-T, which confirms the purity of the produced materials. High-resolution XPS spectra of each element present in the materials (TiO_2_, g-C_3_N_4_ and 30-GCN-T) are presented in Figure 8b–e. Figure 8b shows the comparison between the deconvoluted C 1s core level spectra of g-C_3_N_4_, TiO_2_ and 30-GCN-T. The g-C_3_N_4_ sample was fitted with peaks at 284.8, 286.1 and 288.2 eV of equal full width at half maximum and a broader peak at 293.5 eV. The peaks at 284.8 eV and 286.1 eV are identified as C–C and C–O bonds of adventitious carbon, respectively. The most intense peak at 288.2 eV corresponds to sp^2^ C atoms in the N=C−N aromatic ring of g-C_3_N_4_, whereas the peak at 293.5 eV is related to the three-coordinate C atoms C−NH_2_ [93,94,95]. The C 1s emission of TiO_2_ only presents emissions from adventitious carbon. The 30-GCN-T material presents a mixture of g-C_3_N_4_ and TiO_2_ cases. Since an excellent fit of the 30-GCN-T C 1s emission could be obtained by applying the peak models of both g-C_3_N_4_ and TiO_2_ (with fixed relative binding energies and relative peak areas, respectively), it can be concluded that no significant change occurred in the chemistry of both g-C_3_N_4_ and TiO_2_ when forming the heterostructure. XPS high-resolution O 1s core level spectra for g-C_3_N_4,_ TiO_2_ and 30-GCN-T are also shown in Figure 8c. Only a very small amount of oxygen is present in g-C_3_N_4_, in accordance with the C−O bond of adventitious carbon. For TiO_2_, a typical set of peaks related to Ti−O bonds, undercoordinated oxygen, hydroxyl groups and surface-adsorbed water are found (in ascending order of binding energy) [93,96]. These peaks do not change significantly in their relative areas in a comparison of the TiO_2_ material with the 30-GCN-T material, indicating no chemical change in TiO_2_ upon formation of the heterostructure.

Figure 8d shows the high-resolution Ti 2p core level spectra of TiO_2_ and 30-GCN-T. For TiO_2_, the binding energy values of Ti 2p _3/2_ and Ti 2p _1/2_ obtained at 458.5 and 464.3 eV can be assigned to Ti^4+^ species in the form of TiO_2_ agglomerates [97], and the peak positions are consistent with the reported values for pure TiO_2_ nanostructures [23]. The Ti 2p spectrum of 30-GCN-T is similar to the one of TiO_2_, both in shape and in peak position. Note that it is not possible to comment on eventual binding energy shifts due to the necessary usage of an electron flood gun during the measurement.

Regarding the deconvoluted high-resolution N 1 s spectra of g-C_3_N_4_ and 30-GCN-T (Figure 8e), both materials exhibit five peaks at 398.7, 399.3, 400.0, 401.1, 404.3 eV and 406.6 eV, corresponding to the binding states of sp^2^-hybridized aromatic N atoms bonded to carbon atoms (–C–N=C), tertiary nitrogen N–(C)_3_, C–N–H groups [98], respectively, and two satellite peaks at higher binding energies [99]. No shifts in N 1s spectra were observed.

The N/C ratio of g-C_3_N_4_ was also obtained, and its value was close to the theoretical value of ideal g-C_3_N_4_ (≈1.30). Using the C 1s peak model to determine the carbon of g-C_3_N_4_ inside the 30-GCN-T material also allows quantifying the C/N ratio in 30-GCN-T, which amounts to 1.4, which is very close to pristine g-C_3_N_4_.

### 3.2. Optical Characterization

The optical properties of the produced materials were investigated by UV-VIS absorption and PL spectroscopies (Figure 9). As observed in Figure 9a, pure g-C_3_N_4_ shows an absorption maximum at around 384 nm (3.23 eV), arising from the n → π* transitions caused by the electron transfer from a nitrogen non-bonding orbital to an aromatic anti-bonding orbital [100,101], and an absorption onset at ca. 450 nm. Meanwhile, pure anatase TiO_2_ exhibits an absorption maximum in the UV region and an absorption onset at ca. 390 nm (3.18 eV). The absorption onsets correspond to the optical band gap values (E_g_) of the materials (around 2.8 and 3.2 eV, respectively, for pure g-C_3_N_4_ and pure TiO_2_). These results are in good agreement with the band gap values obtained from DRS spectra by means of the Kubelka–Munk function [102], as depicted in the inset of Figure 9a, and fairly in line with the reported band gap values for these materials [25,103,104,105]. For 30-GCN-T nanostructures, it is clearly seen that two absorption peaks appear: one in the UV region, placed at the same wavelength value as for the absorption maximum of pure TiO_2_, and another one with a lower absorption at ca. 384 nm, most likely related to the presence of g-C_3_N_4_. In addition, compared with pure TiO_2_, these nanostructures exhibited a red shift in the absorption onset, from 390 nm (pure TiO_2_) to around 450 nm, hence suggesting the ability of these materials to absorb in the visible region and a possible enhancement of their photocatalytic activity under visible light.

The PL spectra of TiO_2_, g-C_3_N_4_ and 30-GCN-T materials are also presented in Figure 9b. In the case of pure TiO_2_, barely any emission was detected, while pure g-C_3_N_4_ and 30-GCN-T materials exhibited a similar broad and asymmetric emission band, typical of graphitic carbon nitride [106,107]. The inset of Figure 9b reveals a slight shift in the peak position between the two materials, with the band associated with pure g-C_3_N_4_ peaking at 450 nm (~2.75 eV) while 30-GCN-T exhibiting its maximum at ~446 nm (~2.78 eV). Appendix A depicts the spectral deconvolution of the broad bands for each material using three Gaussian functions. In both cases, the same bands were identified, peaking at 2.817 eV (~440 nm), 2.709 eV (~458 nm) and 2.494 eV (~497 nm), which indicates the contribution of the same optical centers to the overall PL emission. These findings are in line with reports from Yuan et al. and Das et al., who demonstrated that after deconvolution of the g-C_3_N_4_ PL spectrum, three emission centers could be observed in this material at around 431, 458 and 491 nm, originating from the σ*– lone pair (LP), π*–LP and π*–π transition pathways, respectively [108,109]. The slight shift in the peak position observed in the present materials is due to different relative intensities of the deconvoluted bands, specifically, an increase in the relative intensity of the band peaking at the highest energy, while the bands at lower energies experience a reduction in their relative intensity. Although not shown, the same results were observed for 15-GCN-T and 45-GCN-T, attesting to the reproducibility of this behavior. This change in the relative intensity of the PL components can be related to the interaction between the g-C_3_N_4_ nanosheets and the TiO_2_ particles, namely charge transfer phenomena, as reported in other works [110,111], which will be highly beneficial for photocatalytic applications.

### 3.3. Photocatalytic Performance in the Degradation of MO under Solar Simulating Light

The produced nanomaterials in powder form were tested as photocatalysts for the degradation of MO under solar simulating light. The degradation rate was monitored by recording the decay in the absorbance peak intensity at 464 nm [112] using a UV-VIS spectrophotometer in intervals of 30 min (for the first 2 h) and every 1 h up to 240 min (4 h) subsequently (Appendix A). After this time, a transparent solution could be observed (Appendix A). The MO degradation ratio (%) was calculated based on Equation (2):(2)$$\mathrm{Degradation}\;\left(\%\right)=\frac{A_{0}-A}{A_{0}}\times 100$$
where $A_{0}$ is the initial absorbance of the MO solution before irradiation, and A is the absorbance of the MO solution after a certain exposure time (t) [23,78,86,113,114]. Based on Appendix A, it is possible to calculate the ratio between the maximum value of each absorbance spectrum (A) at each exposure time and the initial absorbance of the solution ($A_{0}$), as depicted in Figure 10a. As shown in Figure 10a, MO degradation without the catalyst was insignificant in the dark and under solar simulating light, which indicates that the MO solution is highly stable under light exposure cycles. In fact, a slight increase in MO intensity is observed over time, likely associated with the evaporation of the solvent [86].

MO degradation ratios were obtained through Equation (2), and a minimal MO degradation percentage was obtained (18%) in the presence of pure g-C_3_N_4_ within 4 h under solar simulating light. This phenomenon can be explained by the fast electron–hole recombination in this material, which lowers its photocatalytic activity [115]. In contrast, a substantial increase to 60% of MO degradation was achieved with pure TiO_2_ for the same exposure time.

Many factors influence TiO_2_ photocatalytic behavior, such as the crystalline phase, specific surface area, active facets and particle size [30,116]. Regarding the effect of particle size, large TiO_2_ aggregates composed of nanocrystals were obtained (with an average size of ~543 nm, as confirmed by SEM). It is well known that larger particles possess lower specific surface area and surface-to-volume ratio [86]. In spite of that, some hollow spheres were also observed (Figure 3b), providing more active sites for photocatalytic reactions. In terms of TiO_2_ active surfaces, as revealed by STEM, the (100) surface was exposed in TiO_2_ nanocrystals, which is one of the most active surfaces for photocatalysis. This is due to its superior surface atomic structure (100% five-coordinate Ti atoms (Ti_5c_)), and it possesses a greater electronic band structure, where higher reductive and oxidative photoexcited charge carriers can be generated and transferred to the TiO_2_ surface to react with the pollutant molecules, resulting in higher photocatalytic performance [117,118]. The crystalline phase also plays a key role in photocatalytic activity. Although the synergistic effect of a mixture of TiO_2_ phases seems to be beneficial for photocatalysis, among the different TiO_2_ polymorphs and when it comes to a single phase, TiO_2_ anatase is considered the most photoactive phase. Pure TiO_2_ brookite may present superior photocatalytic activity compared to the other two phases. However, it is difficult to synthesize, making it the less studied TiO_2_ polymorph in photocatalysis [119]. Comparing TiO_2_ rutile and anatase, TiO_2_ rutile suffers from low generation and high recombination rate of charge carriers because of deep electron traps, despite having a lower band gap energy value (~3.0 eV) than TiO_2_ anatase (~3.2 eV). Studies also indicate that TiO_2_ anatase shows a slower charge carrier recombination, since it has the lightest average effective mass of photogenerated electrons and holes, hence the lowest recombination rate of charge carriers [119,120]. In addition, TiO_2_ anatase is considered an indirect band gap semiconductor, thus exhibiting a longer lifetime of photoexcited electrons and holes, since direct transitions of photogenerated electrons from the conduction band to the valence band of anatase are not possible [121].

Notably, all g-C_3_N_4_/TiO_2_ nanostructures exhibited superior photocatalytic performance compared to pure TiO_2_ and pure g-C_3_N_4_ (Figure 10a). As observed previously through the HAADF-STEM image in Figure 4i, an obvious interface was revealed between the TiO_2_ nanocrystal and the g-C_3_N_4_ sheet. Therefore, the enhancement of MO photocatalytic degradation under solar simulating light is expected to be due to an efficient photogenerated charge carriers’ separation at the g-C_3_N_4_–TiO_2_ anatase interface. The 15-GCN-T and 45-GCN-T materials exhibited similar MO photocatalytic degradation percentages (75 and 77%, respectively, in 4 h). The SEM images in Figure 3a show the presence of large TiO_2_ agglomerates in the 15-GCN-T material, leaving some uncovered areas of the g-C_3_N_4_ nanostructures by TiO_2_. The low content of g-C_3_N_4_ may also lead to insufficient visible-light absorption to excite the electrons and holes [122]. With the highest g-C_3_N_4_ content, not all g-C_3_N_4_ nanosheets were covered by TiO_2_ nanoparticles, and thus, this could have induced recombination of photogenerated charges, as previously reported [122].

The material with an intermediate amount of g-C_3_N_4_ (30-GCN-T) demonstrated the best photocatalytic performance, and in the presence of this material, MO degradation of 84% was achieved in 4 h. Therefore, the optimum g-C_3_N_4_ loading in TiO_2_ was found to be 30% in weight.

To determine the MO degradation kinetics, the Langmuir–Hinshelwood model was used, and the simplified pseudo-first-order kinetics equation was applied, as represented in Equation (3).
(3)$$\ln\left(\frac{C}{C_{0}}\right)=-k_{\mathrm{ap}}t$$

In Equation (3), $k_{\mathrm{ap}}$ is the photodegradation apparent rate constant (min^−1^), t is the time (min), $C_{0}$ is the initial concentration (mg/L), and C is the concentration at a certain time (mg/L) [86,114].

The photodegradation apparent rate constants ($k_{\mathrm{ap}}$) were obtained from the plots of $\ln\left(\frac{C}{C_{0}}\right)$ as a function of time (t), as seen in Figure 10b, where the apparent rate constants correspond to the slopes of the linear regressions. Table 2 summarizes the kinetic parameters (rate constants ($k_{\mathrm{ap}}$) and linear regression coefficients R^2^) obtained for the degradation of MO under solar simulating light up to 4 h.

Through the analysis of Table 2, it can be concluded that the photocatalytic dye degradation follows pseudo-first-order kinetics for all synthesized nanostructures, since a good correlation for the fitted lines was obtained (R^2^ > 0.95) [86,114]. A much higher photocatalytic degradation rate in the presence of g-C_3_N_4_/TiO_2_ nanostructures (15-GCN-T, 30-GCN-T and 45-GCN-T) was exhibited compared to the pure materials of TiO_2_ and g-C_3_N_4_. In fact, comparing the rate constant obtained with the best photocatalyst (30-GCN-T), an enhanced efficiency of almost 1.97 and 10 times greater than that of TiO_2_ and g-C_3_N_4_ nanosheets was achieved, respectively. Considering the present results, the improvement of the visible light utilization (Figure 9a), the suppression in the recombination rate of photogenerated charge carriers with respect to g-C_3_N_4_ (Figure 9b) and the interaction between g-C_3_N_4_ and TiO_2_ (Figure 4i and Figure 8) might have helped in boosting the photocatalytic activity in this material.

A direct comparison of related studies, such as the ones presented in Table 1, is not straightforward due to the distinct parameters employed during photocatalytic experiments. Nonetheless, it is noteworthy to mention the simplicity of the method used to prepare g-C_3_N_4_/TiO_2_ heterostructures by using a lower temperature and shorter reaction time.

In future studies, to avoid the drawbacks associated with the recovery and recyclability of powder catalysts, this heterostructure should be synthesized or immobilized on low-cost substrates through a simple and fast approach, such as microwave irradiation, and without the need for using toxic reagents, which has proved to be of great interest in photocatalytic applications [23,86].

#### Reusability Tests and Possible Photocatalytic Degradation Mechanism

Recyclability tests were carried out with the best photocatalyst (30-GCN-T) under solar simulating light along five consecutive cycles of 4 h each. Figure 11a shows the C/C_0_ comparison of photocatalytic MO degradation by 30-GCN-T nanostructures along five cycles. A gradual decrease in the degradation efficiency is observed between cycles, and a degradation loss efficiency of around 20% is obtained at the end of the fifth cycle. A high percentage of MO molecules or reaction products could have remained adsorbed on the surface of the catalyst, retarding the photocatalytic degradation process [123].

The contribution of different ROS to the degradation rate of MO dye using the 30-GCN-T nanostructures was investigated under solar simulating light for 4 h. Although various ROS contribute to the photocatalytic degradation process [78,114], studies have shown that MO degradation is mainly driven by holes (h+), hydroxyl radicals (·OH) and superoxide ions (·O_2_^−^) radicals [124,125]. Therefore, in this study, EDTA, IPA and BQ were used as specific scavengers of holes (h+), hydroxyl radicals (·OH) and superoxide ions (·O_2_^−^). Photocatalytic degradation in the absence and presence of the different scavengers is presented as a MO degradation rate constant (Figure 11b). As shown in Figure 11b, all scavengers inhibited MO degradation. The addition of IPA had little effect on the MO degradation rate, revealing negligible participation of ·OH radicals in the degradation process. Unlike IPA, upon the use of BQ, the removal rate of MO dye by 30-GCN-T was significantly suppressed, thus suggesting that superoxide ions are the primary active species involved in the photocatalytic degradation process. This trend was followed by holes, since the addition of EDTA also showed a significant decrease in the reaction rate. Similar results are also reported in the literature, where superoxide radicals were the main active species, and holes acted as complementary species in MO photodegradation [126].

### 3.4. Electrochemical Characterization

For better insight into the charge transfer process and band alignment of the 30-GCN-T material, the flat band potentials (E_FB_) of TiO_2_ and g-C_3_N_4_ were obtained under 1 kHz from Mott–Schottky plots (M–S plots), as shown in Figure 12. The values were estimated based on the M–S relation (4):(4)$$\frac{1}{C_{\mathrm{int}}^{2}}=\left(\frac{2}{{e\varepsilon \varepsilon }_{0}N_{d}}\right)\times \left(E-E_{\mathrm{FB}}-\frac{\mathrm{KT}}{e}\right)$$
where $C_{\mathrm{int}}^{2}$ is the interfacial capacitance; e is the electron charge; ε is the dielectric constant of the materials; $\varepsilon_{0}$ is the vacuum permittivity (8.85 × 10^−12^ Fm^−1^); $N_{d}$ is the electron donor density; E is the applied potential; $E_{\mathrm{FB}}$ is the flat band potential; K is the Boltzmann constant (1.38 × 10^−23^ JK^−1^); and T is the temperature [79,127]. At RT, $\frac{\mathrm{KT}}{e}$ is negligible [79]. Flat band potentials can be estimated by plotting a graph of 1/$C_{\mathrm{int}}^{2}$ as a function of the potential (V) and can thus be estimated from the intercept of the linear portion of these graphs with the potential axis (*y* axis = 0) [128].

As visible in Figure 12, the positive slope on M–S plot curves suggests that both TiO_2_ and g-C_3_N_4_ materials exhibit an n-type semiconductor behavior [129,130,131]. The flat band potential values against Ag/AgCl were also estimated to be −0.7 V for TiO_2_ and −1.1 V for g-C_3_N_4_. These values are close to the previously reported data under 1 kHz [132,133]. The potentials were recalculated against NHE, according to Equation (1), and values of −0.50 V and −0.90 V were found for TiO_2_ and g-C_3_N_4_, respectively. Moreover, an approximation of the conduction band potential values (E_CB_) can also be performed, whereby for n-type semiconductors, it is often considered that E_FB_ vs. NHE values are more positive at 0.1 V than E_CB_ values [133,134]; therefore, E_CB_ values were determined as −0.6 V and −1 V for TiO_2_ and g-C_3_N_4._ Based on the predicted optical band gap values, as visible in Figure 9a, it is possible to estimate the valence band potential values (E_VB_), which are obtained by adding the E_CB_ values to the optical band gap values [135], resulting in +2.6 V and +1.8 V for TiO_2_ and g-C_3_N_4_, respectively.

Based on the experimental results above, two possible MO degradation mechanisms of the 30-GCN-T material are proposed (Figure 13). Under solar simulating light, electrons are excited from the valence band (VB) of TiO_2_ and g-C_3_N_4_ to the conduction band (CB) and generate electron–hole pairs. Regarding the first mechanism, since the VB potential of g-C_3_N_4_ (+1.8 V vs. NHE) is less positive than the reduction–oxidation potential of OH^−^/·OH (+1.99 V vs. NHE) and H_2_O/·OH (+2.37 V vs. NHE), the holes will not be able to oxidize OH^−^ or H_2_O into ·OH radical species. Nevertheless, a small decrease in the photodegradation percentage of MO was observed previously after the addition of IPA (Figure 11b). Therefore, hydroxyl radicals participated in the degradation process. This was further supported by the decreased percentage of MO degradation after the addition of EDTA because the presence of holes was detected, which could contribute to the generation of ·OH radicals. Therefore, the formation of a Z-scheme mechanism seems possible [136]. In this scheme, the photogenerated electrons in TiO_2_ recombine with the holes in g-C_3_N_4_ through the built-in electrostatic field, resulting in a more efficient separation of photogenerated charge carriers. Due to the more positive VB potential in TiO_2_ (+2.6 V vs. NHE) than that in OH^−^/·OH (+1.99 V vs. NHE) and H_2_O/·OH (+2.37 V vs. NHE) [137], the holes can form hydroxyl radicals by reacting with adsorbed water molecules or surface hydroxyls at the surface of TiO_2_. At the same time, the electrons on the CB of g-C_3_N_4_ could be captured by O_2_ to form ·O_2_^−^ radical species due to the more cathodic CB potential (−1 V vs. NHE) compared to the redox potential of O_2_/·O_2_^−^ (−0.33 V vs. NHE) radicals [137,138]. These hydroxyl and superoxide radical species (OH^−^ and ·O_2_^−^, respectively) will further reduce and oxidize MO dye to H_2_O and CO_2_, with a major contribution from the superoxide radicals, as observed in the photodegradation experiments with ROS (Figure 11b) [139,140]. In this case, the good coupling between TiO_2_ and g-C_3_N_4_ leads to a stronger reduction and oxidation abilities, which will confer improved efficiency to the heterostructure for MO photocatalytic degradation under sunlight.

In contrast to this mechanism, and as revealed before, only a small fraction of hydroxyl radicals was formed during the photocatalytic reaction due to the slight inhibition with IPA. These hydroxyl radicals could have been produced by the reaction of remaining TiO_2_ holes and hydroxyl groups. In this mechanism (type-II heterostructure), electron–hole pairs are produced and separated in both materials upon light irradiation. Photogenerated holes are transferred from TiO_2_ to g-C_3_N_4_ and do not produce hydroxyl radicals, although they might directly oxidize MO dye. Meanwhile, the electrons in g-C_3_N_4_ flow to the conduction band of TiO_2_ and are subsequently scavenged by oxygen on the surface of the catalyst to generate superoxide radicals [111]. Efficient separation of electron–hole pairs is expected, thereby prolonging the lifetime of photogenerated charge carriers, resulting in improvement of the photocatalytic efficiency of the photocatalyst [141].

## 4. Conclusions

In conclusion, visible-light-activated photocatalysts based on g-C_3_N_4_ and TiO_2_ were produced through a simple and fast microwave-assisted approach, contrary to the frequently employed time- and energy-consuming fabrication methods reported in the literature. Different g-C_3_N_4_ amounts in TiO_2_ were investigated, and the produced nanopowders (15-GCN-T, 30-GCN-T and 45-GCN-T) were tested for the degradation of MO under solar simulating light. XRD data indicated the presence of solely anatase TiO_2_ and g-C_3_N_4_ in 30-GCN-T and 45-GCN-T heterostructures, while only the TiO_2_ anatase phase was detected in the 15-GCN-T material, likely due to the low percentage of g-C_3_N_4_ in the heterostructure. The SEM results showed that the use of ethanol as solvent resulted in irregularly shaped TiO_2_ particles, which formed large aggregates. The increase in the amount of g-C_3_N_4_ in TiO_2_ led to the disintegration of these larger particles, and smaller TiO_2_ agglomerates were formed, along with the formation of a TiO_2_ film, which covered the porous g-C_3_N_4_ nanosheets. For the 30-GCN-T material, STEM revealed an established interface between a TiO_2_ crystal and a g-C_3_N_4_ sheet, while XPS confirmed both components to be chemically intact in the heterostructure. The ability to use visible light was demonstrated by the red shift in the absorption onset compared with pure TiO_2_. The best photocatalytic performance was achieved with the 30-GCN-T heterostructure. This heterostructure can degrade 84% of MO dye in 4 h under solar simulating light, corresponding to an enhanced efficiency of almost 2 and 10 times greater than that of TiO_2_ and g-C_3_N_4_ nanosheets, respectively. Two photocatalytic degradation mechanisms were proposed: type-II heterostructure and Z-scheme, where the latter seemed more plausible owing to the small fraction of hydroxyl radical species detected, whereas superoxide radicals were the main active species observed in the ROS scavengers’ experiment. An easy strategy was employed in this study, without the need for pre- or post-treatment processes, in which g-C_3_N_4_/TiO_2_ heterostructures were synthesized through a microwave-assisted solvothermal method with great potential for water decontamination.

## Figures and Tables

**Figure 1 nanomaterials-13-01090-f001:**
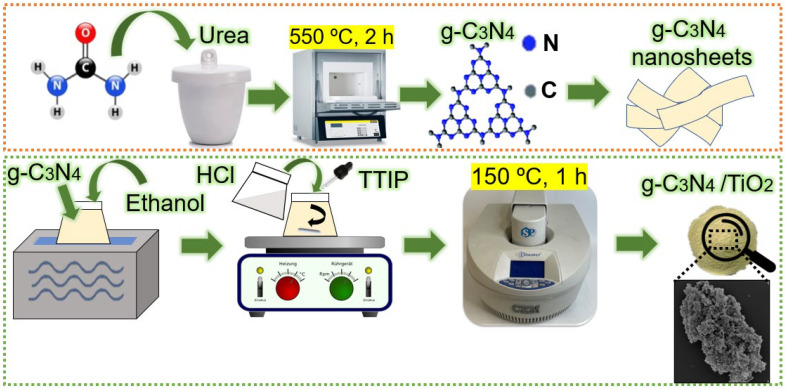
Schematic diagram for the microwave synthesis of g-C_3_N_4_/TiO_2_ heterostructures. The schematic in the orange box represents the first steps for the synthesis of g-C_3_N_4_/TiO_2_ heterostructures, while the schematic in the green box represents the last steps for the synthesis of g-C_3_N_4_/TiO_2_ heterostructures.

**Figure 2 nanomaterials-13-01090-f002:**
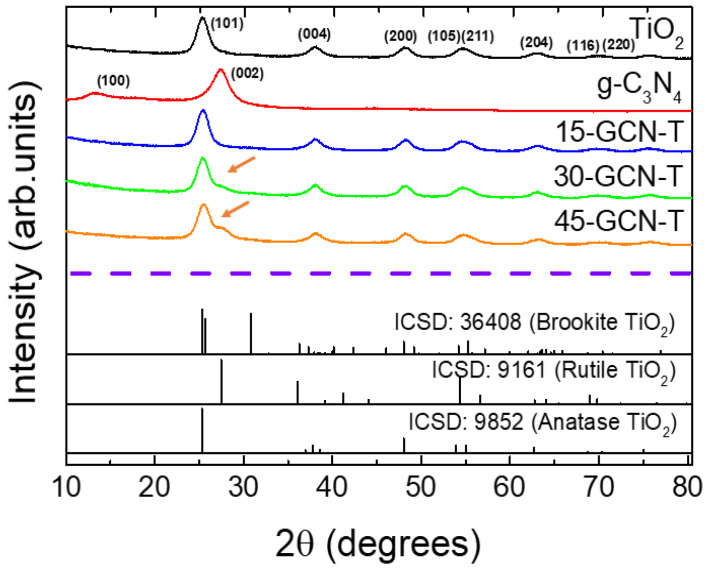
XRD diffractograms of the pure TiO_2_, pure g-C_3_N_4_ and heterostructures composed of TiO_2_ with different weight loading percentages of g-C_3_N_4_ (15-GCN-T, 30-GCN-T and 45-GCN-T). The simulated brookite, rutile and anatase TiO_2_ are also shown for comparison. Orange arrows indicate the diffraction maximum at 27º likely associated with the presence of graphitic carbon nitride.

**Figure 3 nanomaterials-13-01090-f003:**
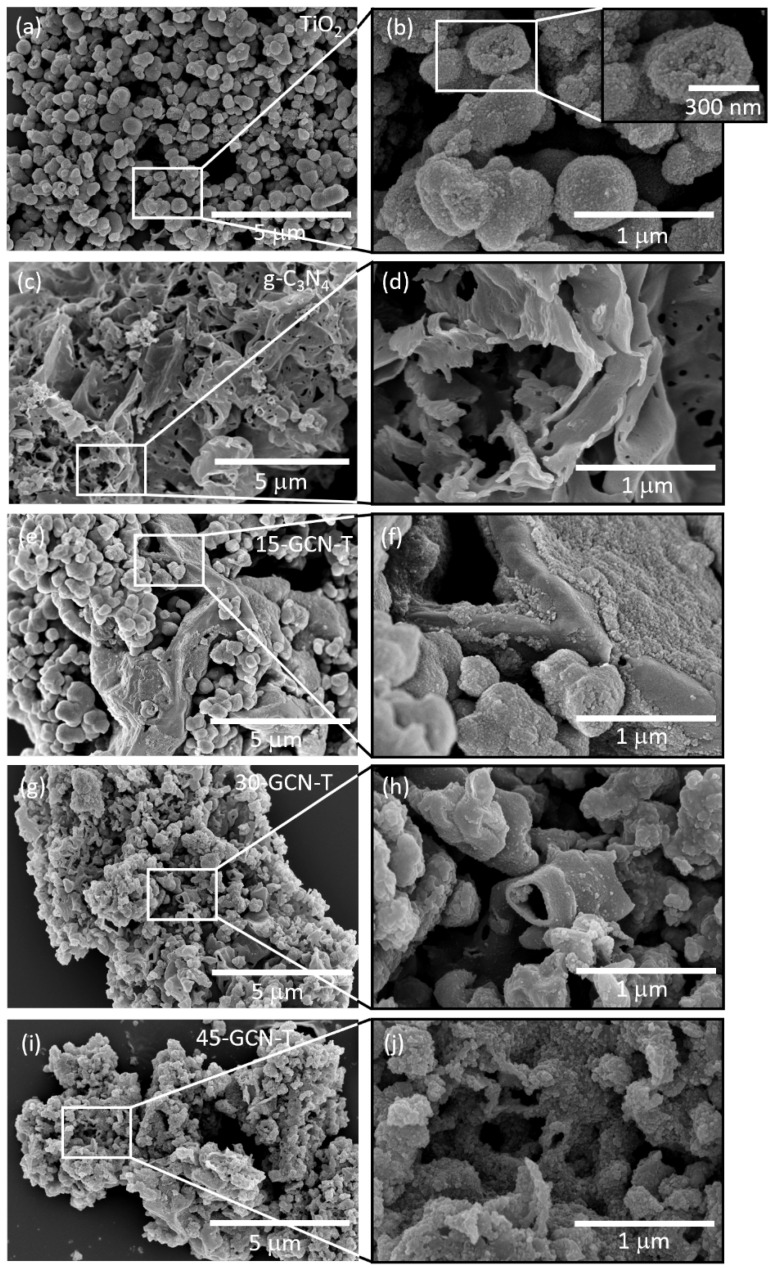
SEM images of the produced materials (**a**) TiO_2_, (**c**) g-C_3_N_4_, (**e**) 15-GCN-T, (**g**) 30-GCT and (**i**) 45-GCN-T. The respective high-magnification SEM images are shown in (**b**,**d**,**f**,**h**,**j**). (**b**) also displays an amplified SEM image of a hollow TiO_2_ sphere.

**Figure 4 nanomaterials-13-01090-f004:**
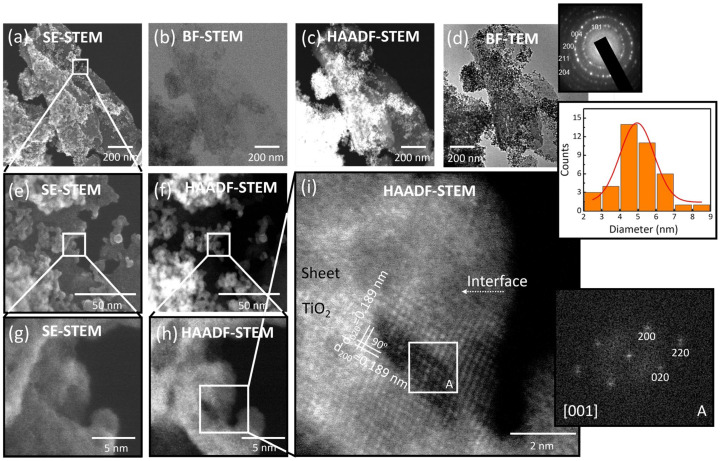
(**a**) Secondary electron (SE) STEM image of the 30-GCN-T material, (**b**) Bright-field (BF) STEM image of the same area of (**a**), (**c**) HAADF-STEM image of the area in (**a**,**d**) Bright-field TEM image of the area in (**a**). The insets in (**d**) depict the electron diffraction pattern of TiO_2_ nanoparticles with the anatase phase and the particle size distribution of the TiO_2_ nanoparticles measured by TEM analyses. (**e**,**f**) Magnified SE-STEM and HAADF-STEM images of the area analyzed in (**a**), respectively. (**g**) SE-STEM and (**h**) HAADF-STEM image of a TiO_2_ nanocrystal attached to the g-C_3_N_4_ sheet, and (**i**) atomic-resolution HAADF-STEM image of the area in (**g**,**h**), where the interface between the TiO_2_ nanocrystal and the g-C_3_N_4_ sheet is clear. The inset in (**i**) shows the FFT image of the area indicated as A (white square).

**Figure 5 nanomaterials-13-01090-f005:**
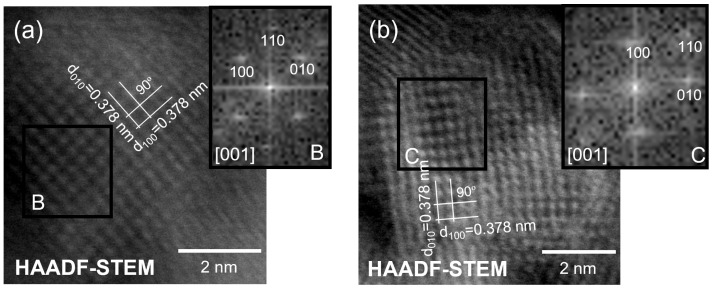
(**a**,**b**) Atomic-resolution HAADF-STEM images of two distinct TiO_2_ nanocrystals. The insets in (**a**,**b**) show the FFT images of areas indicated as B and C, respectively (black squares).

**Figure 6 nanomaterials-13-01090-f006:**
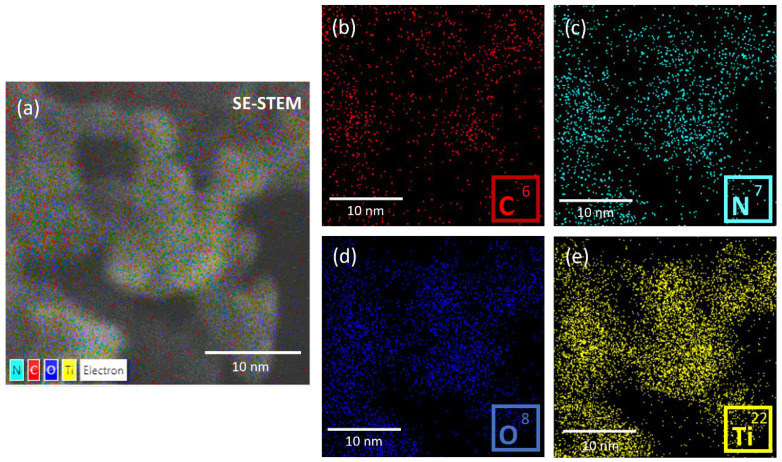
Artificially colored (mixed) SE-STEM image of the 30-GCN-T material (**a**), together with the corresponding EDS maps of C (**b**), N (**c**), O (**d**) and Ti (**e**).

**Figure 7 nanomaterials-13-01090-f007:**
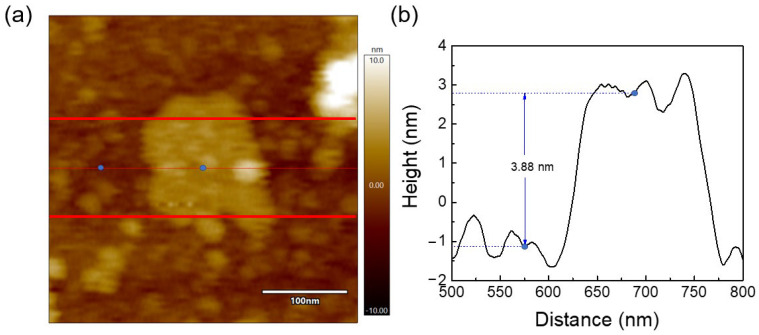
(**a**) AFM image of a g-C_3_N_4_ nanosheet and (**b**) corresponding average height values measured between the red lines.

**Figure 8 nanomaterials-13-01090-f008:**
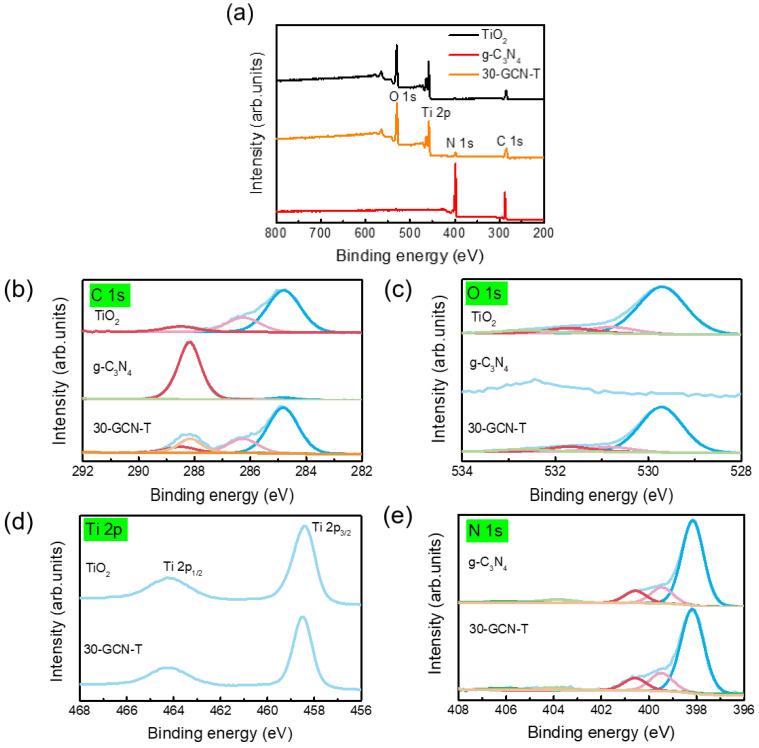
(**a**) XPS survey spectra of TiO_2_, g-C_3_N_4_ and 30-GCN-T materials. (**b**) Deconvolution of XPS C 1s of TiO_2,_ g-C_3_N_4_ and 30-GCN-T spectra. (**c**) Deconvolution of XPS O 1s spectra of TiO_2,_ g-C_3_N_4_ and 30-GCN-T. (**d**) XPS Ti 2p spectra of TiO_2_ and 30-GCN-T. (**e**) Deconvolution of XPS N 1s spectra of g-C_3_N_4_ and 30-GCN-T.

**Figure 9 nanomaterials-13-01090-f009:**
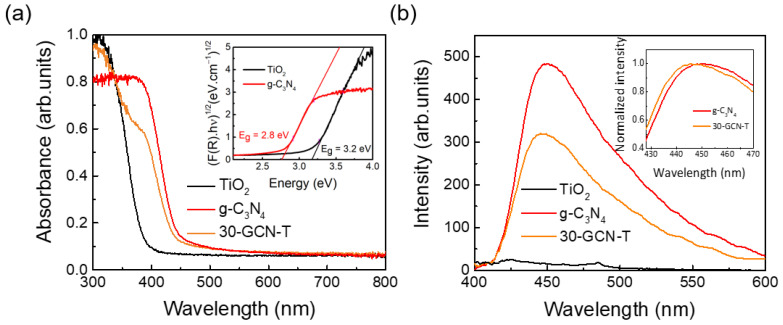
(**a**) RT absorption spectra of TiO_2_, g-C_3_N_4_ and 30-GCN-T nanopowders. The inset shows the Kubelka–Munk plots (from DRS spectra of TiO_2_ and g-C_3_N_4_ nanopowders). For the determination of the optical band gap values, TiO_2_ and g-C_3_N_4_ materials were considered as indirect band gap semiconductors. (**b**) RT PL spectra of TiO_2_, g-C_3_N_4_ and 30-GCN-T nanostructures from 400 to 600 nm using an excitation wavelength of 350 nm. The inset shows the RT PL spectra (normalized intensity as a function of the wavelength from 428 to 470 nm) of g-C_3_N_4_ and 30-GCN-T nanostructures.

**Figure 10 nanomaterials-13-01090-f010:**
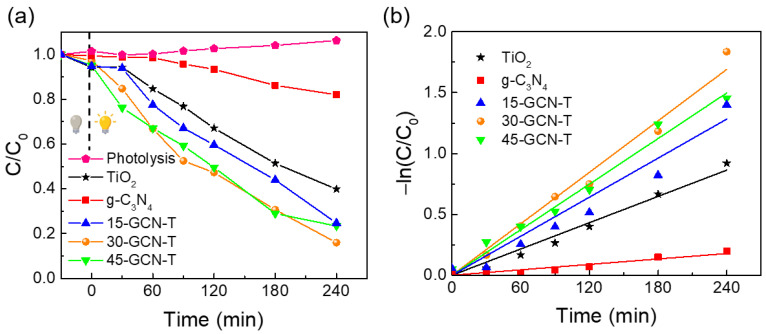
(**a**) Degradation curves (C/C_0_ as a function of the exposure time) under solar simulating light up to 4 h without photocatalyst (photolysis) and for TiO_2_, g-C_3_N_4_, 15-GCN-T, 30-GCN-T,45-GCN-T photocatalysts. The lines are for eye guidance only. (**b**) Pseudo-first-order kinetics for MO degradation in the presence of TiO_2_, g-C_3_N_4_, 15-GCN-T, 30-GCN-T and 45-GCN-T photocatalysts. The lines represent the linear fittings of the pseudo-first-order kinetics equation.

**Figure 11 nanomaterials-13-01090-f011:**
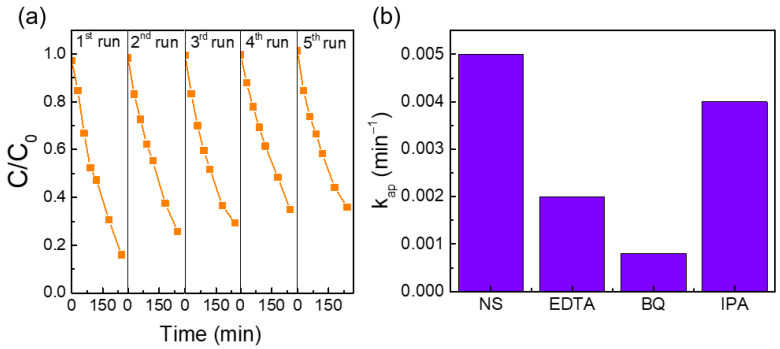
(**a**) MO degradation curves (C/C_0_ as a function of the exposure time) in the presence of 30-GCN-T heterostructures up to 4 h under five consecutive cycles. The lines are for eye guidance only. (**b**) Comparison of the degradation rates of MO under solar simulating light using the 30-GCN-T heterostructure with no scavengers (NS) and 5 mL of water, and in the presence of trapping reagents (EDTA, BQ and IPA).

**Figure 12 nanomaterials-13-01090-f012:**
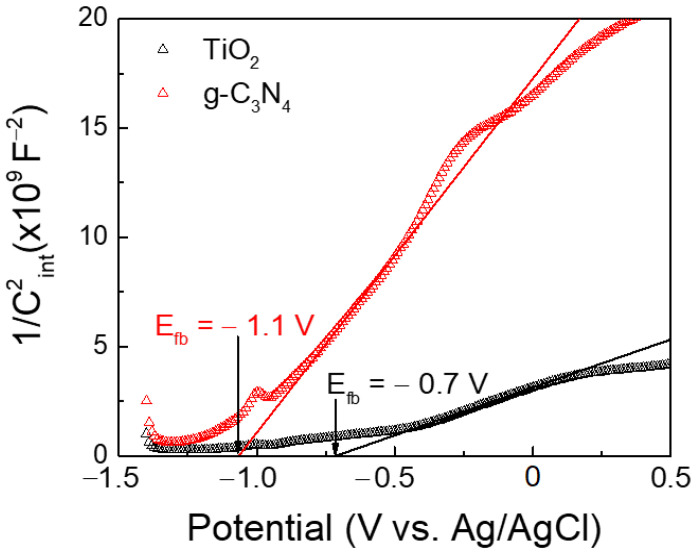
Mott–Schottky analyses performed at 1 kHz in 0.5 M Na_2_SO_4_ electrolyte for TiO_2_ and g-C_3_N_4_. The potentials were measured against the Ag/AgCl reference. The flat band potentials (E_fb_) can be estimated from the linear portion of the graphs, represented in black and red lines, respectively, for TiO_2_ and g-C_3_N_4_.

**Figure 13 nanomaterials-13-01090-f013:**
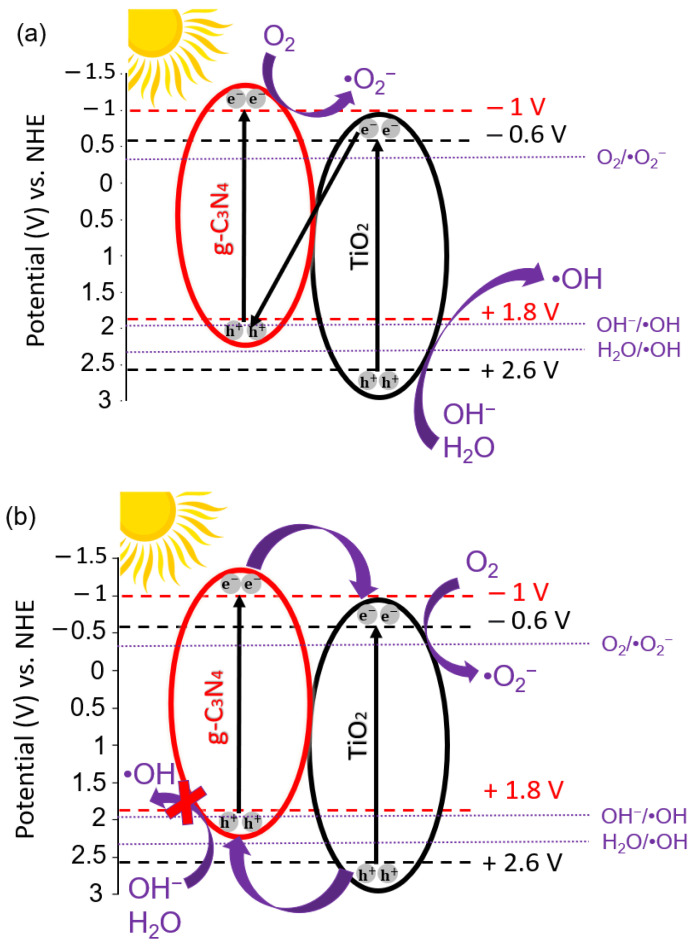
Illustrated mechanism of the photocatalytic activity of the 30-GCN-T heterostructure under solar simulating light. The potentials (V) relative to NHE scale are also represented. Black and red dashed lines correspond to the CB/VB potentials of TiO_2_ and g-C_3_N_4_, respectively. Purple dot lines are related to the redox potentials of the common reactive species in photocatalysis. (**a**) represents a possible Z-scheme photocatalytic degradation mechanism of the 30-GCN-T heterostructure, and (**b**) represents a possible type-II photocatalytic degradation mechanism of the 30-GCN-T heterostructure.

**Table 1 nanomaterials-13-01090-t001:** Summary of g-C_3_N_4_/TiO_2_ nanostructures reported in the literature for the degradation of MO using different preparation methods.

Material	Preparation Method	Light Source	MO Concentration	Optimum Loading	Degradation Efficiency (%)	Kinetic Constant (min^−1^)	Reference
g-C_3_N_4_/seed grown mesoporous TiO_2_	Seed induced solvothermal (MW: 105 °C/48 h)	Visible light	10 mg/L	Ti:g-C_3_N_4_ (1 molar ratio)	Around 100% MO degradation in 60 min (pH = 3)	0.1014	[61]
g-C_3_N_4_/TiO_2_ (brookite)	Calcination (400 °C/1 h)	Visible light	10 mg/L	g-C_3_N_4_:TiO_2_ (35% weight ratio)	55% MO degradation in 180 min	No data	[62]
g-C_3_N_4_/TiO_2_ nanotube array	Anodic oxidation method/ultrasonic loading	Xe lamp irradiation (intensity 100 mW/cm^2^)	15 mg/L	No data	84.6% MO degradation in 120 min	No data	[63]
g-C_3_N_4_ nanosheets/mesoporous TiO_2_	Hydrothermal synthesis (MW: 180 °C/6 h)	300 W Xe lamp irradiation with a cut-off filter (λ > 420 nm)	32.7 mg/L (100 μM)	g-C_3_N_4_:TiO_2_ (2:1 weight ratio)	Around 60% MO degradation in 300 min (pH = 3)	No data	[64]
g-C_3_N_4_ nanosheets/TiO_2_ nanoflakes	in situ sol-gel (400 °C/3 h)	UV-VIS light	20 mg/L	g-C_3_N_4_:TiO_2_ (1:4 weight ratio)	97% MO degradation in 80 min	0.0718	[21]

**Table 2 nanomaterials-13-01090-t002:** Pseudo-first-order kinetic parameters (rate constants and linear regression coefficients) for the photocatalytic degradation of MO under solar simulating light in 4 h by TiO_2_, g-C_3_N_4_, 15-GCN-T, 30-GCN-T and 45-GCN-T nanostructures.

Nanostructures	$k_{ap}\;(\min^{-1})$	R^2^
TiO_2_	0.0036	0.99
g-C_3_N_4_	0.0007	0.95
15-GCN-T	0.0053	0.97
30-GCN-T	0.0071	0.99
45-GCN-T	0.0062	0.98

## Data Availability

The authors confirm that the data supporting the findings of this study are available within the article and its Appendix A.

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
