# Peer review of "Microwave Synthesis of Visible-Light-Activated g-C3N4/TiO2 Photocatalysts"

_nanomaterials, 2023, doi:10.3390/nano13061090_

Round 1

Reviewer 1 Report

Based on a fast (1 h) and simple microwave-assisted approach, Matias et al synthesized g-C3N4/TiO2 heterostructures, which are visible-light activated for the photocatalytic degradation of a recalcitrant azo dye (methyl orange (MO)). It is a significant work with  great potential in water decontamination. However, concerning publication in Nanomaterials, some detailed revision is essential to improve this paper.

1. The authors should emphasize the advantages of this method by showing the differences from other fabrication methods;

2. In the abstract, the expression of “Finally, two possible photocatalytic degradation mechanisms of the heterostructure were proposed” normally is not acceptable for a research article. The authors should say some details for the photocatalytic degradation mechanisms in the abstract to make the readers get a quick overview.

3. Could the author provide an explanation for the experimental result that, the photocatalytic performance of 30-GCN-T is better than the ones of 15-GCN-T and 45-GCN-T?

4. As a work focused on the topic of solving the environmental problems with semiconductors, some current related works should be mentioned in the Introduction, such as Chemical Engineering Journal 410, (2021), 128430; Molecules 28, (2023), 1644; Frontiers of Physics, 15, (2020), 33201; Nanomaterials 11, (2021), 1804; Applied Catalysis B: Environmental, 303, (2022), 120903.

Author Response

  • bold: the reviewer’s comments.
  • normal text: actions taken
  • italics: changes incorporated/that were already included in the manuscript (yellow in the manuscript).

COMMENTS TO THE AUTHOR(S)

Reviewer 1

Based on a fast (1 h) and simple microwave-assisted approach, Matias et al synthesized g-C3N4/TiO2 heterostructures, which are visible-light activated for the photocatalytic degradation of a recalcitrant azo dye (methyl orange (MO)). It is a significant work with  great potential in water decontamination. However, concerning publication in Nanomaterials, some detailed revision is essential to improve this paper.

  1. The authors should emphasize the advantages of this method by showing the differences from other fabrication methods;

  • Authors acknowledge the reviewer’s comment and a paragraph was included in the introduction: Apart from the microwave irradiation process, these methods typically require high temperatures and/or long reaction times, hence being energy-consuming [58,67,68,71,72].”(page 4,lines 136-138)

  1. In the abstract, the expression of “Finally, two possible photocatalytic degradation mechanisms of the heterostructure were proposed” normally is not acceptable for a research article. The authors should say some details for the photocatalytic degradation mechanisms in the abstract to make the readers get a quick overview.
  • Authors acknowledge the reviewer’s comment. The expression “Finally, two possible photocatalytic degradation mechanisms of the heterostructure were proposed” was removed and substituted by “The creation of a type-II heterostructure is highly suggested, due to the negligible participation of hydroxyl radical species in the photodegradation process.”(lines 34-36)

  1. Could the author provide an explanation for the experimental result that, the photocatalytic performance of 30-GCN-T is better than the ones of 15-GCN-T and 45-GCN-T?
  • Authors acknowledge the reviewer’s comment. The explanation was already added. Please refer to page 17, lines 566-569. The low content of g-C3N4 may also lead to insufficient visible-light absorption to excite electrons and holes [123]. For the highest g-C3N4 content, not all g-C3N4 nanosheets were covered by TiO2 nanoparticles and thus could have induced recombination of photogenerated charges, as previously reported [123]”.

  1. As a work focused on the topic of solving the environmental problems with semiconductors, some current related works should be mentioned in the Introduction, such as Chemical Engineering Journal 410, (2021), 128430; Molecules 28, (2023), 1644; Frontiers of Physics, 15, (2020), 33201; Nanomaterials 11, (2021), 1804; Applied Catalysis B: Environmental, 303, (2022), 120903.
  • Authors acknowledge the reviewer’s comment and the suggested references were included: [24], [2],[44], [17] and [56].

Reviewer 2 Report

The rapid synthesis of g-C3N4 photocatalysts are important. The results and findings of this work are interesting and I think it can be accepted after minor revision.

1. The language quality must be improved.

2. The academic writing is not good enough to be published. Do not use abbreviation in abstract.

3. You should have error bar for Figure 10a.

4. Did you test the XRD or SEM of your samples after recycle experiment?

5. The introduction should be polished to show your readers the novelty of your work and some important paper in this field should be referred.

10.26599/NRE.2022.9120015

10.1007/s42765-021-00122-7

10.1016/j.jclepro.2020.125091

10.1007/s42765-022-00192-1

Author Response

  • bold: the reviewer’s comments.
  • normal text: actions taken
  • italics: changes incorporated/that were already included in the manuscript (yellow in the manuscript).

COMMENTS TO THE AUTHOR(S)

Reviewer 2

The rapid synthesis of g-C3N4 photocatalysts are important. The results and findings of this work are interesting and I think it can be accepted after minor revision.

  1. The language quality must be improved.
  • Authors acknowledge the reviewer’s comment and the whole manuscript was reviewed.

  1. The academic writing is not good enough to be published. Do not use abbreviation in abstract.
  • Authors acknowledge the reviewer’s comment. Authors reviewed the manuscript and all abbviations have their meaning explained in the abstract.

  1. You should have error bar for Figure 10a.
  • Authors acknowledge the reviewer’s comment, but at the present moment the microwave equipment is not operational and waiting for techincal intervention. Therefore, we cannot produce more nanopowders to be re-tested.

  1. Did you test the XRD or SEM of your samples after recycle experiment?
  • Authors acknowledge the reviewer’s comment. In fact, the XRD and SEM measurements for the 30-GCN-T sample were carried out after recycling tests. No changes could be detected before and after the photocatalytic experiments using both techniques, and for that reason the results were not worthen of being included.

  1. The introduction should be polished to show your readers the novelty of your work and some important paper in this field should be referred.

10.26599/NRE.2022.9120015   

10.1007/s42765-021-00122-7 

10.1016/j.jclepro.2020.125091

10.1007/s42765-022-00192-1

  • Authors acknowledge the reviewer’s comment and the suggested references were included: [49], [59],[60] and [8].

Reviewer 3 Report

Review of the manuscript „ Microwave synthesis of visible-light activated g-C3N4/TiO2 photocatalysts“ by M.L. Matias, A.S. Reis-Machado, J. Rodrigues, T. Calmeiro, J. Deuermeier, A. Pimentel, E. Fortunato, R. Martins and D. Nunes  (MDPI)

 The paper deals with research related to the application of microwave synthesis for the preparation of visible light-activated g-C3N4-TiO2 photocatalyst, detailed characterization, and activity testing of the thus prepared composite for the degradation of azo dye (MO) under simulated solar irradiation. The work is quite extensive and contains interesting results. The possibility of direct use of sunlight is a state- of- the art in the field of photocatalysis. The topic is very interesting and appealing. The paper is written in a clear, easy-to-read and understandable way. The interpretation of the obtained results is done in an acceptable way. From a technical point of view, I have a serious objection to this work, which concerns the use of azo dye as a model component. In fact, there are thousands of papers in the literature related to the photodegradation of different types of dyes. However, there are many very persistent and priority compounds that certainly deserve the attention of researchers in the field of photocatalysis. Since the authors have nicely and comprehensively presented the obtained results on the mentioned topic, I support the publication of their work in the journal Nanomaterials with a minor modification.

- The authors should explain the increase of normalised concentrations (C/C0) as a function of irradiation time during photolysis (Fig. 10).

- It is necessary to critically analyze the kinetic parameters obtained and compare them with the results published by other authors (e.g., those from Table 1).
- Persulfate activation is also a promising advanced oxidation process for the removal of pollutants for water purification. Have the authors considered the possibility of using PDS or PMS to enhance the photocatalytic degradation of the model compound (MO)?

In conclusion, I recommend acceptance of this manuscript for publication in the journal Nanomaterials after minor changes.

Author Response

  • bold: the reviewer’s comments.
  • normal text: actions taken
  • italics: changes incorporated/that were already included in the manuscript (yellow in the manuscript).

COMMENTS TO THE AUTHOR(S)

Reviewer 3

 The paper deals with research related to the application of microwave synthesis for the preparation of visible light-activated g-C3N4-TiO2 photocatalyst, detailed characterization, and activity testing of the thus prepared composite for the degradation of azo dye (MO) under simulated solar irradiation. The work is quite extensive and contains interesting results. The possibility of direct use of sunlight is a state-of-the-art in the field of photocatalysis. The topic is very interesting and appealing. The paper is written in a clear, easy-to-read and understandable way. The interpretation of the obtained results is done in an acceptable way. From a technical point of view, I have a serious objection to this work, which concerns the use of azo dye as a model component. In fact, there are thousands of papers in the literature related to the photodegradation of different types of dyes. However, there are many very persistent and priority compounds that certainly deserve the attention of researchers in the field of photocatalysis. Since the authors have nicely and comprehensively presented the obtained results on the mentioned topic, I support the publication of their work in the journal Nanomaterials with a minor modification.

  1. The authors should explain the increase of normalised concentrations (C/C0) as a function of irradiation time during photolysis (Fig. 10).
  • Authors acknowledge the reviewer’s comment. The explanation was added in page 16, lines 515-517: “In fact, a slight increase in MO intensity is observed over time, likely associated with the evaporation of the solvent [87].”

  1. It is necessary to critically analyze the kinetic parameters obtained and compare them with the results published by other authors (e.g., those from Table 1).

Authors acknowledge the reviewer’s comment.  The comparison was made in the Results and Discussion section, page 18, lines 603-607: The direct comparison of related studies such as the ones presented in Table 1 is not straightforward due to the distinct parameters employed during photocatalytic experiments, nonetheless, it is noteworthy to mention the simplicity of the method used to prepare g-C3N4/TiO2 heterostructures by using a lower temperature and shorter reaction time”.

  1. Persulfate activation is also a promising advanced oxidation process for the removal of pollutants for water purification. Have the authors considered the possibility of using PDS or PMS to enhance the photocatalytic degradation of the model compound (MO)?

In conclusion, I recommend acceptance of this manuscript for publication in the journal Nanomaterials after minor changes.

  • Authors acknowledge the reviewer’s comment. According to the literature, PS and PMS do not easily react with organic pollutants and need activation to produce organic active species (https://doi.org/10.3389/fchem.2020.592056). Despite that, the method has shown encouraging results in the field of wastewater treatment. In the future, it could be indeed interesting to explore persulfate activation and compare the results with the traditional photocatalytic experiments.
